# Fluorescent and Mechanical Properties of Silicon Quantum Dots Modified Sodium Alginate-Carboxymethylcellulose Sodium Nanocomposite Bio-Polymer Films

**DOI:** 10.3390/polym11091476

**Published:** 2019-09-09

**Authors:** Yali Ji, Huimin Zhang, Canfu Zhang, Zhiyi Quan, Min Huang, Lili Wang

**Affiliations:** 1College of Chemistry, Chemical Engineering and Resource Utilization, Northeast Forestry University, Harbin 150040, China; 2Ministry of Education Key Laboratory of Synthetic and Natural Functional Molecule Chemistry, College of Chemistry & Materials Science, Northwest University, Xi’an 710127, China

**Keywords:** biopolymer, silicon quantum dots, nanocomposite polymer films, fluorescent, mechanical property

## Abstract

Highly luminescent silicon quantum dots (SiQDs) were prepared via one-pot hydrothermal route. Furthermore, the optimal synthetic conditions, dependence of the emission spectrum on the excitation wavelength and fluorescent stability of SiQDs were investigated by fluorescence spectroscopy. SiQDs exhibited bright blue fluorescence, and photoluminescence (PL) lifetime is 10.8 ns when excited at 325 nm. The small-sized SiQDs (~3.3 nm) possessed uniform particle size, crystal lattice spacing of 0.31 nm and silicon (111), (220) crystal planes. Luminescent SiQDs/sodium alginate (SA)-carboxymethylcellulose sodium (CMC) nanocomposite bio-polymer films were successfully fabricated by incorporating SiQDs into the SA-CMC matrix. Meanwhile, SiQDs not only impart strong fluorescence to the polymer, but also make the composite films have favorable toughness.

## 1. Introduction

Due to the quantum confinement effect [1,2], quantum dots are of great interest in applications of light and electricity [3,4,5]. In particular, luminescent silicon quantum dots (SiQDs), as representative zero-dimensional silicon nanostructures, are promising for solar cells [6], light emitting diodes [7], photocatalytic degradation [8], bio-imaging [9] and fluorescent probes [10] with low cytotoxicity [11], excellent water dispersibility [12], favorable biocompatibility [13], surface modification [11,14] and so on. Although, preparation methods of SiQDs have been extensively explored with electrochemical etching silicon wafers using HF [15], reduction of hydrogen silsesquioxane by H_2_ at high temperature [16], laser pyrolysis of silanes [17], microwave assisted method for reducing silane coupling agent by sodium citrate [18], gas phase reduction method for reducing SiO_2_ gel by H_2_ or Mg vapor [19], liquid phase reduction silane method for lithium aluminum hydride reduction halosilane [20], high energy grinding ball method based on silicon wafer [21] and so on, these synthetic methods possess complicated procedures, hazardous reagents (HF, H_2_, H_2_SO_4_, SiX_4_), complex morphology, low purity, uneven size and highly demanding equipment. However, one-pot hydrothermal method [22,23] significantly enhanced synthetic range of SiQDs, shortens the reaction time, and also obtained the particles with a uniform structure.

Recently, many composites have been reported and have attracted considerable attention. In particular, the introduction of polymers not only provides mechanical and chemical stability for nanocomposites, but also reduces the aggregation of quantum dots to maintain their emission properties [24,25]. In the reported literature, poly(vinyl alcohol) (PVA)/GQD and carboxymethyl chitosan/ZnOQD nanocomposite films exhibit toilless biodegradability, excellent biocompatibility, edibility and non-toxicity [26,27]. Sodium alginate (SA) and carboxymethylcellulose sodium (CMC) were chosen as matrix polymers because of high optical transparency, chemical stability, film-forming properties, and abundant and sustainable sources [28,29]. Therefore, in this work, SiQDs were synthesized via one-pot hydrothermal method with ethanol as reagent, silicic acid and sodium borohydride as new raw materials. Furthermore, to obtain a luminescent SA-CMC hybrid film, SiQDs and SA-CMC mixed polymer matrix are poured on the glass plate, and the film is dried after static defoaming. The obtained SiQDs/SA-CMC nanocomposites are promising in optoelectronic application, e.g., light emitting diodes have been reported recently [30].

## 2. Materials and Methods

### 2.1. Materials

Sodium borohydride (99%) was purchased from Tianjin Damao Chemical Reagent Co. Ltd. (Tianjin, China). Silicic acid (99%) was purchased from Aladdin reagent Co. Ltd. (Shanghai, China). Ethanol (99%) and ethyl acetate (99%) were purchased from Tianjin Fuyu Fine Chemical Co. Ltd. (Tianjin, China). Sodium alginate (SA; 99%) was purchased from Tianjin Fuchen Chemical Reagent Co. Ltd. (Tianjin, China). Carboxymethylcellulose sodium (CMC; 99%) was purchased from Tianjin Kermiou Chemical Reagent Co. Ltd. (Tianjin, China). Glycerol (99%) was purchased from Tianjin Zhiyuan Chemical Reagent Co., Ltd. (Tianjin, China). All the reagents and chemicals were of analytical grade without further purification. Deionized water (Milli-Q Academic, Beijing, China) was used throughout the whole experiment. Disposable needle filter (13 μm × 0.22 μm) was purchased from Jiangsu Green Union Science Instrument Co. Ltd. (Jiangsu, China).

### 2.2. Preparation of SiQDs

Firstly, silicic acid of 0.39 g (5 mmol) and ethanol (8 mL) was put into a reactor. Sodium borohydride of 0.227 g (6 mmol) was slowly added to the above system under stirring for 10 min during normal temperature and pressure. Secondly, the reactor was placed in an oven at 200 °C for 3 h. Thirdly, the prepared sample was naturally cooled and ultra-filtered (aperture: 0.22 μm) to obtain a light yellow solution. Then, resulting silicon quantum dots were dried in the oven at 40 °C and were labeled as SiQDs.

### 2.3. Fabrication of SiQDs/SA-CMC Nanocomposites

The SA, CMC and glycerol concentrations were 1.2 g/100 mL, 2.8 g/100 mL and 2 g/100 mL, respectively, and the SA/CMC blend matrix was prepared by stirring for 6 h at 80 °C. Then SiQDs and SA-CMC aqueous solution were mixed. The mixtures were cast onto glass substrates with no post treatment. Furthermore, the glass substrates were transferred to the oven and dried for 12 h at 45 °C to obtain SiQDs/SA-CMC films (40 square centimeters). The SiQDs/SA-CMC composites with 6, 12 and 18 wt% loading of SiQDs were fabricated, respectively. Thus, the resulting SiQDs/SA-CMC films were labeled as 6 wt% SiQDs/SA-CMC, 12 wt% SiQDs/SA-CMC and 18 wt% SiQDs/SA-CMC, respectively.

### 2.4. Characterization Techniques

The morphology was examined in a TECNAI 10 PHILIPS transmission electron microscope (TEM; Amsterdam, Holland) at the accelerating voltage of 100 kV. The powder X-ray diffraction (XRD) pattern of the samples was recorded by a D/MAX 2200 diffractometer (Rigaku, Tokyo, Japan) using Cu Kα radiation in a scan step of 0.02° and a scan range between 10° and 70°. The Fourier transform infrared (FTIR) spectroscopy was performed in ethanol solvent by an Avatar 360 Fourier transform infrared spectrometer (Thermo Nicolet Co., Waltham, MA, USA) using air as the background in three scan times and a scan range between 400 cm^−1^ and 4000 cm^−1^. The X-ray photoelectron spectroscopy (XPS) was tested with THERMO-X-ray photoelectron spectrometer (Thermo Electron Co., Waltham, MA, USA). The photoluminescence spectra (PL) of SiQDs (the concentration in ethanol solvent was 20%) and SiQDs/SA-CMC nanocomposites were recorded on a LS55 fluorescence spectrophotometer (Perkin Elmer Co., USA) using 365 nm excitation in a scan speed of 500 nm/s and slit width of 8 nm, respectively. PL lifetime decay of SiQDs (the concentration in ethanol solution was 20%) was examined with F-7000 fluorescence spectrophotometer (Japan). The strain–stress curves were tested with the LRX-PLUS universal material testing machine (Aiteng Co., Shanghai, China) using SiQDs/SA-CMC sample (length of 8 cm, width of 1 cm and thickness of 0.05 cm) in a scan speed of 500 mm/min.

## 3. Results and Discussion

### 3.1. PL Analysis of SiQDs Under Different Reaction Conditions

To obtain optimally synthetic conditions of SiQDs, the ratio of the reactants, reaction time and temperature were investigated. The effect of molar ratio of reactants on the fluorescence intensity of SiQDs was explored by fixing the reaction temperature at 200 °C and reaction time at 2 h. Figure 1a shows PL spectra of SiQDs with different molar ratio of reactants, the molar ratio of NaBH_4_ to H_2_SiO_3_ was 0.3, 0.6, 0.9, 1.2, 1.5, 1.8 and 3.0. As the molar ratio of NaBH_4_ to H_2_SiO_3_ was from 0.3 to 0.9, the change in fluorescence intensity of SiQDs was increasing, as the molar ratio of NaBH_4_ to H_2_SiO_3_ increased to 1.2:1, the fluorescence intensity was maximum. However, as the molar ratio of NaBH_4_ to H_2_SiO_3_ was from 1.5 to 3.0, the fluorescence intensity weakened. The reason was that excessive NaBH_4_ would lead to partial fluorescence quenching of SiQDs, which was caused by concentration quenching. Therefore, when the molar ratio of NaBH_4_ to H_2_SiO_3_ was 1.2:1, the reductant NaBH_4_ reduced the silicon source H_2_SiO_3_ to form silica-oxygen cross-linking core and the formation of silica-oxygen cross-linking core could be well promoted. The bonding of SiQDs forms a local trap state in the broadened bandgap, which transfers more electrons from the occupied orbit to the new local trap state [22]. At this time, the fluorescence intensity was the highest, that is, the molar ratio of the best reactant.

The reaction temperature was investigated by fixing reaction time at 2 h and the molar ratio of the reactants at 1.2. Figure 1b shows the fluorescence spectra of SiQDs at 180 °C, 190 °C, 200 °C, 210 °C and 220 °C, respectively. When the reaction temperature reached 200 °C, SiQDs reached the highest fluorescence intensity due to reduction of surface defects, generation of silicon-oxygen crosslinking core and shape of new gap states. However, after 200 °C, the agglomeration of SiQDs caused the weakening of the fluorescence intensity. Therefore, the optimum reaction temperature for the preparation of SiQDs was 200 °C.

The effect of reaction time on the preparation of SiQDs was investigated by fixing the molar ratio of NaBH_4_ to H_2_SiO_3_ at 1.2:1 and the reaction temperature at 200 °C. Figure 1c shows the fluorescence spectra of SiQDs with reaction time from 1 to 8 h. The fluorescence intensity of SiQDs increased firstly and then decreased. The reason for the decrease of fluorescence intensity was that SiQDs tend to agglomerate with the increase of reaction time. Therefore, the optimum reaction time for preparing SiQDs was 3 h.

Figure 1d shows the fluorescence spectra of SiQDs at different excitation wavelengths under optimally synthetic conditions with the molar ratio of NaBH_4_ to H_2_SiO_3_ at 1.2:1, reaction temperature at 200 °C and reaction time at 3 h. When the excitation wavelength increased from 355 to 425 nm, the fluorescence intensity of SiQDs show the gradual decrease and red shift of the emission wavelength. The result indicates that the emission wavelength of SiQDs had a strong dependence on the excitation wavelength. Figure 1d confirmed that the emission peak ranged from 400 to 500 nm, belonging to the blue light range. Blue fluorescence was also confirmed by 365 nm ultraviolet lamp irradiation. Its digital photograph showed the inset of Figure 1d. The fluorescence spectra of Figure 1a–c all confirm that the characteristic emission wavelength of SiQDs was 440 nm.

### 3.2. Morphology Analysis of SiQDs

Figure 2 illustrates the TEM image and particle size distribution of SiQDs with optimally synthetic conditions, respectively. From Figure 2a, it can see that SiQDs were spherical with good dispersion and small size. The enlarged high-resolution transmission electron microscope (HRTEM) had exhibited in the inset of Figure 2a. SiQDs possess excellent crystallinity with a well-resolved (111) lattice spacing of 0.31 nm, which was consistent with the reported literature [14]. Figure 2b displays the size distribution histogram measured from 100 particles. The size of most SiQDs concentrated at 3 to 4 nm, a small portion was at 1 to 2 nm and the average particle size was about 3.3 nm.

### 3.3. XRD and FTIR Analysis of SiQDs

Figure 3a displays the XRD pattern of SiQDs with optimally synthetic conditions. From Figure 3a, it can be observed that the dominant peaks of SiQDs appear at 2θ values of 28.9° and 47.5°, corresponding to (111) and (220) planes, respectively. Figure 3b shows the FTIR spectroscopy of SiQDs. The bands at 650 cm^−1^ and 1068 cm^−1^ belonged to the symmetrical and antisymmetrical stretching vibrations of Si–O–Si, respectively. The peak at 860 cm^−1^ originated from the stretching vibration of Si–C. The bands of 1397 cm^−1^ and 1435 cm^−1^ stood for the deformation vibrations of –CH_3_ and –CH_2_ groups. The bands of 2900 cm^−1^ and 2962 cm^−1^ were anti-symmetric stretching vibration of –CH_2_ and–CH_3_, respectively, indicating that the prepared SiQDs were attached by C–H_X_ groups. The peak at 3320 cm^-1^ was attributed to the stretching vibration of the –OH group.

### 3.4. XPS Analysis of SiQDs

Figure 4 shows the XPS spectra of SiQDs with optimally synthetic conditions. From Figure 4a, the binding energy of Si 2p at 99.40 eV and 101.9 eV were attributed to Si–Si and Si–C bonds, respectively. Since the peak intensity of zero-valent silicon (Si–Si) was lower than that of tetravalent silicon (Si–C), it was mainly tetravalent silicon in SiQDs, which was consistent with the reports in literature [31]. Figure 4b shows the XPS spectrum of O 1s. The binding energy at 532 eV indicates the existence of Si–O, while the small peak at 536 eV indicates that the prepared SiQDs had not only Si–O, but also Si–OH. The XPS spectrum of C 1s in Figure 4c shows that the binding energy of C–Si at 284 eV, which also proved that tetravalent silicon mainly existed in SiQDs. The small peak at 287 eV was attributed to C–H_x_, which was consistent with the infrared analysis. In general, Si–O–Si, Si–C and Si–OH groups play a key role in the stability of SiQDs.

### 3.5. Fluorescence Stability Analysis of SiQDs

The fluorescent spectra of SiQDs with optimally synthetic conditions were measured every 5 days with a total of 35 days, and the maximum PL intensity is illustrated in Figure 5a. The fluorescent intensity decreased from 1 to 10 days because of the agglomeration of SiQDs during the testing process [32]. However, SiQDs with a stable structure remained fluorescently stable after 10 days. Notably, the SiQDs exhibited superior photostability compared to fluorescein isothiocyanate (FITC) dye, CdTe and CdSe/ZnS QDs, put as the fluorescent labels of photostability [33]. In particular, the fluorescence of FITC, CdTe QDs and CdSe/ZnS QDs was strong at the beginning, but gradually decreased as the exposure time increased. Figure 5b shows the PL lifetime decay curves of SiQDs. Lifetime of the resultant SiQDs was 10.8 ns, much longer than FITC dye (3.80 ns) and R6G (4.3 ns). Furthermore, the long-lifetime fluorescence of SiQDs was easily distinguished from the autofluorescence of the living organism, whose lifetimes were less than 5 ns [9]. Such SiQDs were not only promising in optoelectronic application, but significance of imaging application in vitro and vivo.

### 3.6. Formation Mechanism of SiQDs

Through the above analysis of FTIR and XPS, SiQDs possessed a Si–O–Si bond, Si–Si bond and Si–C bond, thus the growth mechanism of SiQDs was explained by these bonds, as shown in Figure 6. In the hydrothermal process, the growth of SiQDs followed the bottom-up synthesis mechanism, that is, the bond first forms a core in hydrophilic groups such as –OH, and then the core forms a QD via the Ostwald ripening process [23]. Under high temperature and pressure, Si–O–Si bonds, Si–Si bonds and Si–C bonds first form Si–O–Si cores, Si cores and Si–C cores in the hydrophilic environment (–OH groups of ethanol). Then the above cores grow gradually to form steady SiQDs that undergo the Ostwald ripening process, respectively.

### 3.7. PL Analysis of SiQDs/SA-CMC Nanocomposites

Figure 7a shows the PL spectra of SA-CMC nanocomposites with SiQDs loadings of 6 wt%, 12 wt% and 18 wt% at 365 nm excitation wavelength. The characteristic emission wavelength of SiQDs/SA-CMC nanocomposites appears at 440 nm, which is consistent with that of SiQDs. Figure 7b shows the maximum emission intensity of SiQDs/SA-CMC nanocomposite films. The emission intensity increased linearly with the increase of SiQDs content. It could be interpreted that luminescent properties were significantly enhanced, due to the exciton formation, that is, the charge trapping effect: QDs can trap electrons and allow more holes to recombine through the interface of polymers and QDs [34,35]. The inset in Figure 7b is a digital photograph of SiQDs/SA-CMC nanocomposite films in a dark box under 365 nm UV lamp irradiations. Compared with the inset of Figure 1d, the fluorescence color of the nanocomposite film was the same as that of the SiQDs. Moreover, the fluorescence brightness of the composite film increased with the increase of SiQDs content.

### 3.8. Mechanical Properties Analysis of SiQDs/SA-CMC Nanocomposites

Figure 8 exhibits strain–stress curves of SA-CMC and SiQDs/SA-CMC. Young’s modulus of materials is the ratio of stress to strain within the elastic limit. Young’s modulus represents the stiffness of the material, the larger the Young’s modulus, the harder to deform. Young’s modulus of pure SA-CMC and SiQDs/SA-CMC with SiQDs loadings of 6 wt%, 12 wt% and 18 wt% were 0.0125, 0.0055, 0.0053 and 0.0036 MPa, respectively. The results indicate that the stiffness of nanocomposites decreased with the addition of SiQDs.

Strain at break is also named elongation, which is utilized to characterize the ductility and toughness of polymer materials. The strain at break of SA-CMC and SiQDs/SA-CMC with SiQDs loadings of 6 wt%, 12 wt% and 18 wt% was 300%, 325%, 350% and 390%, respectively. The increase of strain at break demonstrates that the toughness of SA-CMC film could be improved through adding SiQDs. The ultimate stress of pure SA-CMC and SiQDs/SA-CMC with SiQDs loadings of 6 wt%, 12 wt% and 18 wt% was 1.64, 1.40, 1.60 and 1.67 MPa, respectively, the above data changed little.

The fracture energy was the area under the stress–strain curve. It could be also indicated the toughness of polymer materials. The area under the stress–strain curve of SA-CMC and SiQDs/SA-CMC with SiQDs loadings of 6 wt%, 12 wt% and 18 wt% was 300, 348, 415 and 487 N/m, respectively. The increase of fracture energy under the stress–strain curve indicates that the toughness of SA-CMC film could be improved through adding SiQDs. Furthermore, the higher the ultimate stress or strain at break, the greater fracture energy, thus it was consistent with the situation where the elongation at break increased and the ultimate stress changed little in this experiment.

It could be interpreted that the decreasing Young modulus, increasing strain at break and fracture energy and improving toughness of SA-CMC composites with the addition of SiQDs, was because the interfacial adhesion was better and the effective interface was enhanced between SiQDs and the SA-CMC matrix when SiQDs was added into SA-CMC composites. Meanwhile, due to the existent Si–O–Si bonds in SiQDs, sols were formed in solution easily, and the sol in water could interact with SA-CMC to form a colloidal particles penetrating SA-CMC network structure.

## 4. Conclusions

In summary, SiQDs were synthesized via a simple one-pot hydrothermal method. The results indicate that SiQDs had a characteristic emission wavelength at 440 nm, and the emission spectrum had strong dependence on the excitation wavelength. In addition, emission intensity increased linearly. Strain at break of SA-CMC film enhanced and Young’s modulus reduced through adding SiQDs. More significantly, this new kind of SiQDs/SA-CMC nanocomposites might show high optical applications, owing to low cost, non-toxicity, excellent fluorescence and favorable toughness.

## Figures and Tables

**Figure 1 polymers-11-01476-f001:**
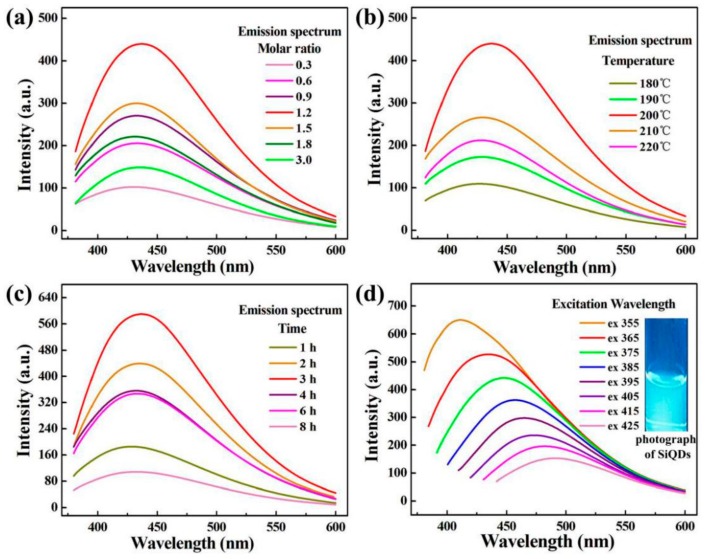
Photoluminescence (PL) spectra of silicon quantum dots (SiQDs). (**a**) Different molar ratios; (**b**) different temperatures; (**c**) different reaction times (λex = 365 nm) and (**d**) different excitation wavelengths (λex = 355–425 nm). Inset in (**d**) is digital photographs of SiQDs under 365 nm UV lamp irradiations.

**Figure 2 polymers-11-01476-f002:**
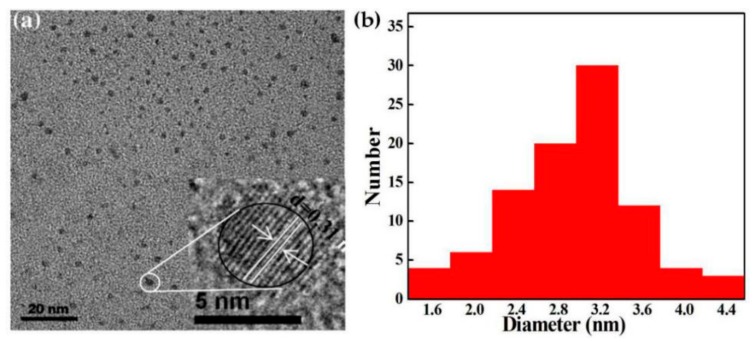
(**a**) TEM image and (**b**) size-distribution of SiQDs.

**Figure 3 polymers-11-01476-f003:**
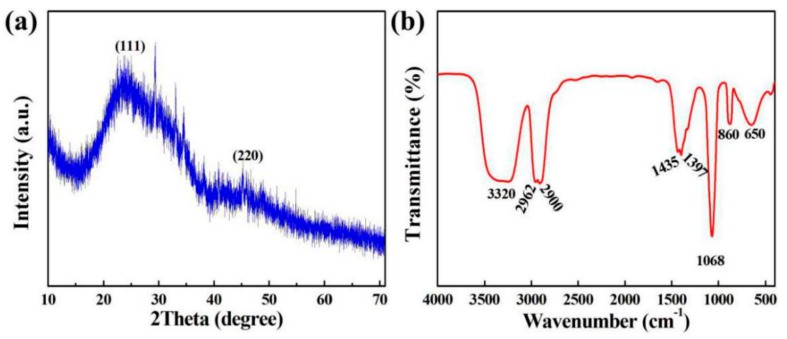
(**a**) XRD pattern of SiQDs and (**b**) FTIR spectrum of SiQDs.

**Figure 4 polymers-11-01476-f004:**
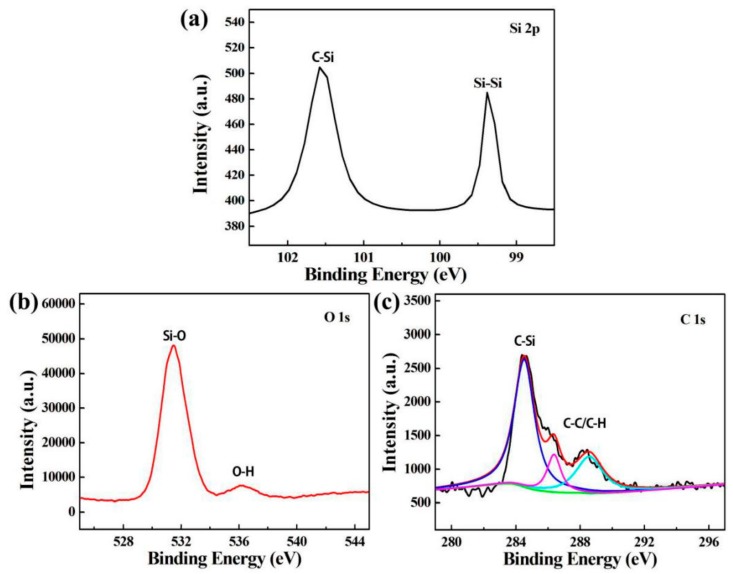
XPS spectra of (**a**): Si (2p) (**b**): O (1s) and (**c**) C (1s).

**Figure 5 polymers-11-01476-f005:**
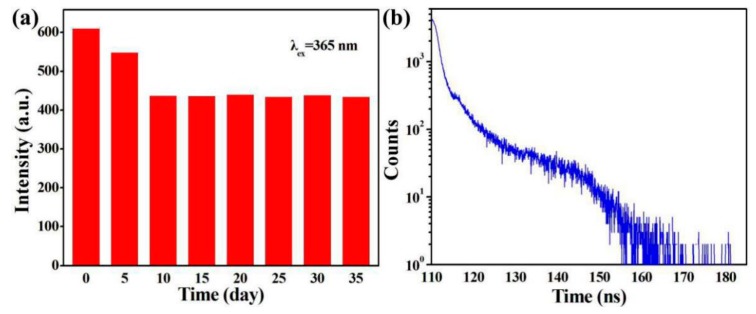
(**a**) Maximum intensity of PL spectra and (**b**) PL lifetime decay curves of SiQDs.

**Figure 6 polymers-11-01476-f006:**
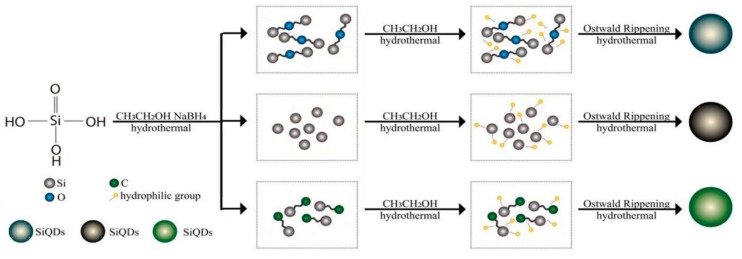
Mechanism of SiQDs.

**Figure 7 polymers-11-01476-f007:**
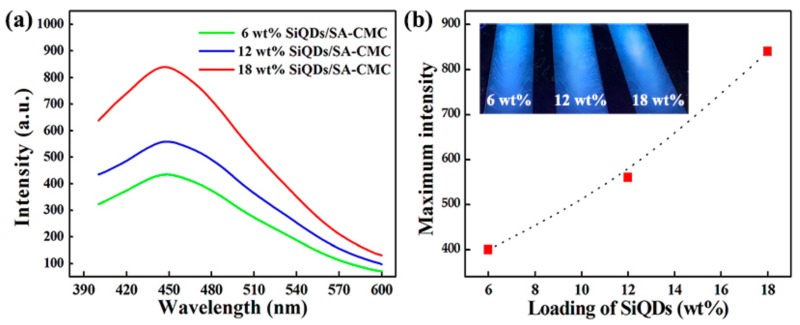
(**a**) PL spectra (ex = 365 nm) and (**b**) maximum intensity of SiQDs/SA-CMC nanocomposites. Inset in (**b**) is digital photographs of SiQDs/SA-CMC nanocomposite films under 365 nm UV lamp irradiations.

**Figure 8 polymers-11-01476-f008:**
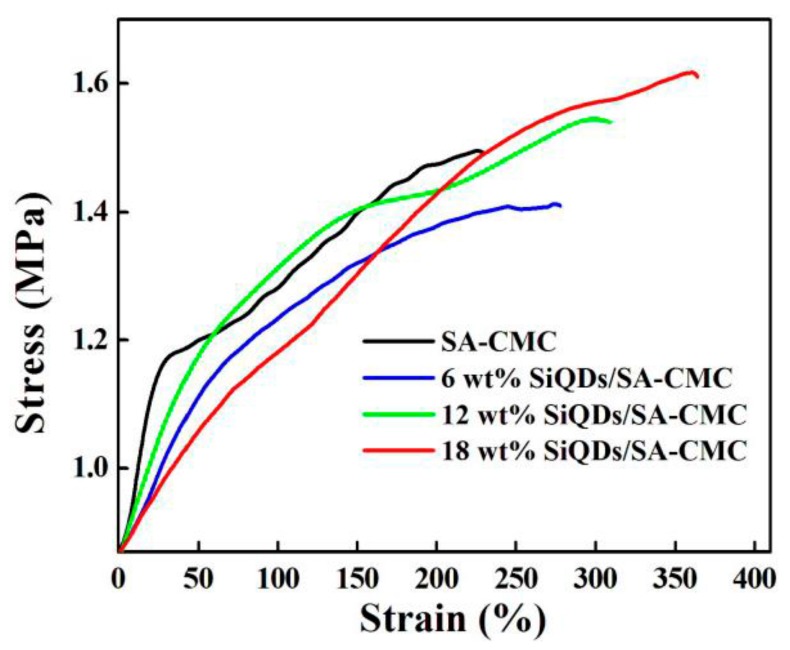
Strain–stress curves of pure SA-CMC and SiQDs/SA-CMC nanocomposites.

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
