# Peer review of "Fluorescent and Mechanical Properties of Silicon Quantum Dots Modified Sodium Alginate-Carboxymethylcellulose Sodium Nanocomposite Bio-Polymer Films"

_polymers, 2019, doi:10.3390/polym11091476_

Round 1

Reviewer 1 Report

Dear authors,

I read your manuscript and here are my comments:

Which is the novelty of your paper with respect to existent literature?

Mechanical tests: you should explain why the Young modulus decreases with the addition of SiQDs. The same should be explained for elongation at break increase.

Materials section: Sodium alginate (CMC)  - I think it is SA. You should correct the error.

Which are the MWs of CMC and SA? This is important for their solubility, especially for SA that could easily form gels.

It is better to provide concentrations for SA and CMC solutions instead of 1.2 g and 2.8 g.

What about the surface characteristics of your films such as roughness or contact angle? Did you perform contact angle measurements?

Which is the target application for these nanocomposite films?

Author Response

Response to Reviewer 1 comments

Firstly I would like to take this opportunity to thank you for your carefully reading and your constructive comments and suggestions.

Answer to Reviewer #1:

Q1-1: Which is the novelty of your paper with respect to existent literature?

A1-1. Answer to reviewer: 

Thanks you for your question.

The following points are the novelty of this paper with respect to existent literature: 1) In this work, SiQDs were firstly synthesized via one-pot hydrothermal method with ethanol as reagent, silicic acid and sodium borohydride as new raw materials. 2) To obtain a luminescent SA-CMC hybrid film, SiQDs were firstly introduced into the SA-CMC polymer matrix through a solution casting method. 3) Aggregation and high cost of SiQDs can result in limited application and slow development, therefore, the SiQDs/SA-CMC nanocomposites are promising in optoelectronic application (such as light emitting diodes). 4) The SA-CMC matrix film forming solution possesses the advantages of toilless biodegradability, excellent biocompatibility, edibility and non-toxicity.

The relevant literatures are as follows:

15.He, Y.; Zhong, Y.; Peng, F.; Wei, X.; Su, Y.; Lu, Y.; Su, S.; Gu, W.; Liao, L.; Lee, S.T. One-pot microwave synthesis of water-dispersible, ultraphoto- and pH-stable, and highly fluorescent silicon quantum dots. J. Am. Chem. Soc. 2011, 133, 14192-14195. (see Literature [15])

16.Qian, C.; Sun, W.; Wang, L.; Chen, C.; Liao, K.; Wang, W.; Jia, J.; Hatton, B.D.; Casillas, G.; Kurylowicz, M.; Yip, C.M.; Mastronardi, M.L.; Ozin, G.A. Non-wettable, oxidation-stable, brightly luminescent, perfluorodecyl-capped silicon nanocrystal film. J. Am. Chem. Soc. 2014, 136, 15849-15852. (see Literature [16])

17.Stanca, L.; Petrache, S.N.; Serban, A.I.; Staicu, A.C.; Sima, C.; Munteanu, M.C.; Zărnesc, O.; Dinu, D.; Dinischiotu, A. Interaction of silicon-based quantum dots with gibel carp liver: oxidative and structural modifications. Nanoscale Res. Lett. 2013, 8, 254-264. (see Literature [17])

30.Han, Y.; Wang, L. Sodium alginate/carboxymethyl cellulose films containing pyrogallic acid: physical and antibacterial properties. J. Sci. Food Agric. 2017, 97, 1295-1301. (see Literature [30])

31.Sathesh, V.; Chen, J.-K.; Chang, C.-J.; Aimi, J.; Chen, Z.-C.; Hsu, Y.-C.; Huang, Y.-S.; Huang, C.-F. Synthesis of Poly(ε-caprolactone)-Based Miktoarm Star Copolymers through ROP, SA ATRC, and ATRP. Polymers. 2018, 10, 858. (see lines Literature [31])

The relevant literatures added are as follows:

18.Wang, J.J.; Lian, M.; Xiong, J. A new water-soluble imidazole based silicon quantum dot was prepared and used for fluorescence detection of trace copper in fruits and vegetables. J. Chin. Anal. Chem. 2016, 44, 367. (see Literature [18])

19.Anderson, I.E.; Shircliff, R.A.; Macauley, C. Silanization of Low-Temperature-Plasma Synthesized Silicon Quantum Dots for Production of a Tunable, Stable, Colloidal Solution. J. Phys. Chem. C. 2012, 116, 3979. (see Literature [19])

20.Bley, R.A.; Kauzlarich, S.M. A low-temperature solution phase route for the synthesis of silicon nanoclusters. J. Am. Chem. Soc. 1996, 118, 12461. (see Literature [20])

21.Heintz, A.S.; Fink, M.J.; Mitchell, B.S. Mechanochemical synthesis of blue luminescent alkyl/alkenyl-passivated silicon nanoparticles. Adv. Mater. 2007, 19, 3984. (see Literature [21])

26.Kovalchuk, A.; Huang, K.; Xiang, C.; Marti, A.A.; Tour, J.M. Luminescent Polymer Composite Films Containing Coal-Derived Graphene Quantum Dots. Acs Appl. Mater. Interfaces 2015, 7, 26063-26068. (see Literature [26])

27.Li, T.; Li, B.; Ji, Y.; Wang, L. Luminescent and UV-Shielding ZnO Quantum Dots/Carboxymethylcellulose Sodium Nanocomposite Polymer Films. Polymers 2018, 10, 1112-1119. (see Literature [27])

Q1-2: Mechanical tests: you should explain why the Young modulus decreases with the addition of SiQDs. The same should be explained for elongation at break increase.

A1-2. Answer to reviewer: 

We appreciate your suggestion very much.

We have added the content of fracture energy and explained the reason that Young modulus decreases and elongation at break increase with the addition of SiQDs in the revised manuscript. “The fracture energy is the area under stress-strain curve. It could be also indicated the toughness of polymer materials. The area under stress-strain curve of SA-CMC and SiQDs/SA-CMC with SiQDs loadings of 6 wt%, 12 wt% and 18 wt% is 300, 348, 415 and 487 N/m, respectively. The increase of fracture energy under stress-strain curve indicates that the toughness of SA-CMC film can be improved through adding SiQDs. Furthermore, the higher ultimate stress or strain at break, the greater fracture energy, thus it is consistent with the situation where the elongation at break increases and ultimate stress changes little in this experiment.”

“It can be interpreted that the decreasing Young modulus, increasing strain at break and fracture energy, and improving toughness of SA-CMC composites with the addition of SiQDs, because the interfacial adhesion is better and the effective interface enhances between SiQDs and the SA-CMC matrix when SiQDs was added into SA-CMC composites. Meanwhile, due to the existent Si-O-Si bonds in SiQDs, sols are formed in solution easily, and the sol in water can interact with SA-CMC to form a colloidal particles penetrating SA-CMC network structure.”

 The previous content “Figure 8 exhibits strain-stress curves of SA-CMC and SiQDs/SA-CMC. The ultimate stress of pure SA-CMC and SiQDs/SA-CMC with SiQDs loadings of 1, 2 and 3 ml is 1.64, 1.40, 1.60 and 1.67 MPa, respectively. In comparison with pure SA-CMC film, the ultimate stress of 1 mL-SiQDs/SA-CMC and 2 mL-SiQDs/SA-CMC reduces and 3 mL-SiQDs/SA-CMC slightly increases. Ultimate stress reflects the fracture resistance of materials, the larger ultimate stress, the stiffer material. The data above exhibits quite small, therefore, it can be confirmed that SiQDs/SA-CMC films are ductile material and SiQDs has little influence on ultimate stress of pure SA-CMC.”

“Young's modulus of materials is the ratio of stress to strain within the elastic limit. Young's modulus represents the stiffness of the material, the larger Young's modulus, the harder to deform. Young's modulus of pure SA-CMC and SiQDs/SA-CMC with SiQDs loadings of 1, 2 and 3 ml are 0.0125, 0.0055, 0.0053 and 0.0036 MPa, respectively. The results indicate that the Young's modulus of nanocomposites decrease with the addition of SiQDs.”

“Strain at break is also named elongation, which is utilized to characterize the toughness or brittleness of polymer materials. The strain at break of SA-CMC and SiQDs/SA-CMC with SiQDs loadings of 1, 2 and 3 ml is 300%, 325%, 350% and 390%, respectively. The increase of strain at break demonstrates that the flexibility of SA-CMC film can be improved through adding SiQDs. ” has been changed into “Figure 8 exhibits strain-stress curves of SA-CMC and SiQDs/SA-CMC. Young's modulus of materials is the ratio of stress to strain within the elastic limit. Young's modulus represents the stiffness of the material, the larger Young's modulus, the harder to deform. Young's modulus of pure SA-CMC and SiQDs/SA-CMC with SiQDs loadings of 6 wt%, 12 wt% and 18 wt% are 0.0125, 0.0055, 0.0053 and 0.0036 MPa, respectively. The results indicate that the stiffness of nanocomposites decrease with the addition of SiQDs.

Strain at break is also named elongation, which is utilized to characterize the ductility and toughness of polymer materials. The strain at break of SA-CMC and SiQDs/SA-CMC with SiQDs loadings of 6 wt%, 12 wt% and 18 wt% is 300%, 325%, 350% and 390%, respectively. The increase of strain at break demonstrates that the toughness of SA-CMC film can be improved through adding SiQDs. The ultimate stress of pure SA-CMC and SiQDs/SA-CMC with SiQDs loadings of 6 wt%, 12 wt% and 18 wt% is 1.64, 1.40, 1.60 and 1.67 MPa, respectively, the above data change little.

The fracture energy is the area under stress-strain curve. It could be also indicated the toughness of polymer materials. The area under stress-strain curve of SA-CMC and SiQDs/SA-CMC with SiQDs loadings of 6 wt%, 12 wt% and 18 wt% is 300, 348, 415 and 487 N/m, respectively. The increase of fracture energy under stress-strain curve indicates that the toughness of SA-CMC film can be improved through adding SiQDs. Furthermore, the higher ultimate stress or strain at break, the greater fracture energy, thus it is consistent with the situation where the elongation at break increases and ultimate stress changes little in this experiment.

It can be interpreted that the decreasing Young modulus, increasing strain at break and fracture energy, and improving toughness of SA-CMC composites with the addition of SiQDs, because the interfacial adhesion is better and the effective interface enhances between SiQDs and the SA-CMC matrix when SiQDs was added into SA-CMC composites. Meanwhile, due to the existent Si-O-Si bonds in SiQDs, sols are formed in solution easily, and the sol in water can interact with SA-CMC to form a colloidal particles penetrating SA-CMC network structure.”. (see lines 220-244)  

Q1-3: Materials section: Sodium alginate (CMC)  - I think it is SA. You should correct the error.

A1-3. Answer to reviewer: 

Thank you for your correction, it is my honor to be pointed out the writing error.

We have corrected it in the revised manuscript. The previous words “Sodium alginate (CMC)” has been changed into “Sodium alginate (SA)”. The revised sentence is as follows: “Sodium alginate (SA) (99%) was purchased from Tianjin Fuchen Chemical Reagent Co. Ltd. (Tianjin, China).” (see lines 62-63).

Q1-4: Which are the MWs of CMC and SA? This is important for their solubility, especially for SA that could easily form gels.

A1-4. Answer to reviewer: 

Thank you for your question. CMC and SA MWs are 43.54×104 and 54.74×104.

Q1-5: It is better to provide concentrations for SA and CMC solutions instead of 1.2 g and 2.8 g.

A1-5. Answer to reviewer: 

We appreciate your suggestion very much. 

The previous content “SA (1.2 g) and CMC (2.8 g) were completely dissolved in 100 mL of distilled water with stirring at 80 °C.” has been changed into “The SA, CMC and glycerol concentrations were 1.2g/100 mL, 2.8g/100 mL and 2g/100 mL, respectively, and the SA/CMC blend matrix was prepared stirring for 6 h at 80 °C.” in the revised manuscript. (see lines 77-78)  

Q1-6: What about the surface characteristics of your films such as roughness or contact angle? Did you perform contact angle measurements?

A1-6. Answer to reviewer: 

Thank you for your question.

As for surface characteristics, we did not perform contact angle measurements, but the prepared film has a smooth surface without cracks. The reason is that adding glycerin reduces the wrinkle of the CMC-SA composite film after drying. We would provide the kind of photograph for the prepared film, as shown in following photograph.

Q1-7: Which is the target application for these nanocomposite films?

A1-7. Answer to reviewer: 

We appreciate your question very much.

Aggregation and high cost of SiQDs can result in limitedly optoelectronic application and slow development, therefore, we would display the application with respect to existent literature: “The SiQDs/SA-CMC nano-composites are promising in optoelectronic application, e.g. light emitting diodes have been reported recently.” (see lines 55-56)

The relevant literature is as follows:

 Wu, Y.S.; Zhang, H.; Pan, A.Z.; Wang, Q.; Zhang, Y.F.; Zhou, G.J.; He, L. White-Light-Emitting Melamine-Formaldehyde Microspheres through Polymer-Mediated Aggregation and Encapsulation of Graphene Quantum Dots. Adv. Sci. 2019, 6. (seeLiterature [32])

As for the responses to Reviewer 1, it should be mentioned that the revised words are in blue color in the revised manuscript. 

Reviewer 2 Report

The manuscript by Ji et al describes the synthesis of silicon quantum dots at different conditions and their use in biocomposite materials. The manuscript is well written and study is relevant in the field of natural-based polymer composites. Therefore, publication can be recommended after some minor revision.

The methods used should described better. In section 2.2., one example of reaction conditions is give. Authors could add for example table where all the conditions are listed. When making the nanocomposites, the concentration of SiQDs in the solution should be given. Also, the area of the glass substrate, that is, area of the films should be given. Characterization should also be described better, e.g. what was the concentration of SiQDs in the solution during spectroscopy measurements.

Toughness is the area under stress-strain curve, so the strain (or elongation at break) itself is not characteristic to the toughness, i.e. material can have high strain, but if the strength is not high, the toughness can be low. Therefore, authors should calculate the actual toughness to see if it really changes when SiQDs are added. In addition, there is something wrong in the stress-strain curves, as the SA-CMC curve start above 1 MPa and 1ml- SiQDs/SA-CMC at around 10% strain.

It would be good that, based on the properties of nanocomposites, authors could give some ideas where this type of materials could be used.

Author Response

Response to Reviewer 2 comments

Firstly I would like to take this opportunity to thank you for your carefully reading and your constructive comments and suggestions.

Answer to Reviewer #2:

Q2-1: The methods used should described better. In section 2.2., one example of reaction conditions is give. Authors could add for example table where all the conditions are listed.

A2-1. Answer to reviewer: 

We appreciate your suggestion very much.

As for example table where all the conditions are listed in section 2.2, I will put forward a little opinion, sincerely. We only analyzes and lists all the reaction conditions in section 3.1 of results and discussion, but involves an optimally synthetic conditions singly in other sections of results and discussion. Therefore, in section 2.2, we add for example table where all the conditions are listed, the expression of the article will be repeated and confused. The above content only represents personal opinions. If it is unreasonable, we will add for example table where all the conditions are listed in section 2.2 in the revised manuscript. In addition, we have listed all the reaction conditions in section 3.1 analysis of results and discussion in the revised manuscript: “The effect of molar ratio of reactants on the fluorescence intensity of SiQDs was explored by fixing the reaction temperature at 200 °C and reaction time at 2 h. Figure 1a shows PL spectra of SiQDs with different molar ratio of reactants, the molar ratio of NaBH4 to H2SiO3 is 0.3, 0.6, 0.9, 1.2, 1.5, 1.8 and 3.0.” (see lines 104-107) “The reaction temperature was investigated by fixing reaction time at 2 h and molar ratio of the reactants at 1.2. Figure 1b shows the fluorescence spectra of SiQDs at 180 °C, 190 °C, 200 °C, 210 °C and 220 °C, respectively.” (see lines 118-120) “The effect of reaction time on the preparation of SiQDs was investigated by fixing the molar ratio of NaBH4 to H2SiO3 at 1.2:1 and the reaction temperature at 200 °C. Figure 1c shows the fluorescence spectra of SiQDs with reaction time from 1 to 8 h.” (see lines 125-127)  

Q2-2: When making the nanocomposites, the concentration of SiQDs in the solution should be given. Also, the area of the glass substrate, that is, area of the films should be given.

A2-2. Answer to reviewer: 

Thank you for your question.

1) The previous content “Then SA-CMC of 10 mL and SiQDs with 1, 2 and 3 mL mixed solutions were cast onto glass substrates, respectively. The above SiQDs/SA-CMC nanocomposites transferred to the oven and dried at 45 °C for 12 h. The obtained SiQDs/SA-CMC nanocomposite films were labeled as 1 mL-SiQDs/SA-CMC, 2 mL-SiQDs/SA-CMC and 3 mL-SiQDs/SA-CMC, respectively.” has been changed into “Then SiQDs and SA-CMC aqueous solution were mixed. The mixtures were cast onto glass substrates with no post treatment. Furthermore, the glass substrates were transferred to the oven and dried for 12 h at 45 °C to obtain SiQDs/SA-CMC films (40 square centimeters). The SiQDs/SA-CMC composites with 6, 12 and 18 wt% loading of SiQDs were fabricated, respectively. Thus, the resulting SiQDs/SA-CMC films were labeled as 6 wt% SiQDs/SA-CMC,12 wt% SiQDs/SA-CMC and 18 wt% SiQDs/SA-CMC, respectively.” in the revised manuscript. (see lines 78-84);

2) The previous content “Figure 7a shows the PL spectra of SA-CMC nanocomposites with SiQDs loadings of 1, 2 and 3 ml at 365 nm excitation wavelength.” has been changed into “Figure 7a shows the PL spectra of SA-CMC nanocomposites with SiQDs loadings of 6 wt%, 12 wt% and 18 wt% at 365 nm excitation wavelength.” in the revised manuscript. (see lines 204-205);

 3) The previous content “Young's modulus of pure SA-CMC and SiQDs/SA-CMC with SiQDs loadings of 1, 2 and 3 ml are 0.0125, 0.0055, 0.0090 and 0.0036 MPa, respectively.” has been changed into “Young's modulus of pure SA-CMC and SiQDs/SA-CMC with SiQDs loadings of 6 wt%, 12 wt% and 18 wt% are 0.0125, 0.0055, 0.0053 and 0.0036 MPa, respectively.” in the revised manuscript. (see lines 223, 231 and 234).

Q2-3: Characterization should also be described better, e.g. what was the concentration of SiQDs in the solution during spectroscopy measurements.

A2-3. Answer to reviewer: 

We appreciate your suggestion very much. 

The concentration of SiQDs in the solution was 20% during PL measurements. The previous content “The photoluminescence spectra (PL) of SiQDs and SiQDs/SA-CMC nanocomposites were recorded on a LS55 fluorescence spectrophotometer (Perkin Elmer Co., USA), respectively. The photoluminescence spectra (PL) of SiQDs and SiQDs/SA-CMC nanocomposites were recorded on a LS55 fluorescence spectrophotometer (Perkin Elmer Co., USA), respectively.” has been changed into “The photoluminescence spectra (PL) of SiQDs (the concentration in solution was 20%) and SiQDs/SA-CMC nanocomposites were recorded on a LS55 fluorescence spectrophotometer (Perkin Elmer Co., USA) using 365 nm excitation in a scan speed of 500 nm/s and slit width of 8 nm, respectively. PL lifetime decay of SiQDs (the concentration in solution was 20%) was examined with F-7000 fluorescence spectrophotometer (Japan).” in the revised manuscript. (see lines 93-98)  

Q2-4: Toughness is the area under stress-strain curve, so the strain (or elongation at break) itself is not characteristic to the toughness, i.e. material can have high strain, but if the strength is not high, the toughness can be low. Therefore, authors should calculate the actual toughness to see if it really changes when SiQDs are added. In addition, there is something wrong in the stress-strain curves, as the SA-CMC curve start above 1 MPa and 1ml- SiQDs/SA-CMC at around 10% strain.

A2-4. Answer to reviewer: 

Thank you for your question.

As for toughness, we would put forward a little opinion, sincerely. Area under stress-strain curve is fracture energy, the higher ultimate stress and strain at break, the greater fracture energy, thus it is consistent with the elongation at break ultimate stress. We have added the content of fracture energy and corrected the toughness in  the revised manuscript. “The fracture energy is the area under stress-strain curve. It could be also indicated the toughness of polymer materials. The area under stress-strain curve of SA-CMC and SiQDs/SA-CMC with SiQDs loadings of 6 wt%, 12 wt% and 18 wt% is 300, 348, 415 and 487 N/m, respectively. The increase of fracture energy under stress-strain curve indicates that the toughness of SA-CMC film can be improved through adding SiQDs. Furthermore, the higher ultimate stress or strain at break, the greater fracture energy, thus it is consistent with the situation where the elongation at break increases and ultimate stress changes little in this experiment.”

The previous content “Figure 8 exhibits strain-stress curves of SA-CMC and SiQDs/SA-CMC. The ultimate stress of pure SA-CMC and SiQDs/SA-CMC with SiQDs loadings of 1, 2 and 3 ml is 1.64, 1.40, 1.60 and 1.67 MPa, respectively. In comparison with pure SA-CMC film, the ultimate stress of 1 mL-SiQDs/SA-CMC and 2 mL-SiQDs/SA-CMC reduces and 3 mL-SiQDs/SA-CMC slightly increases. Ultimate stress reflects the fracture resistance of materials, the larger ultimate stress, the stiffer material. The data above exhibits quite small, therefore, it can be confirmed that SiQDs/SA-CMC films are ductile material and SiQDs has little influence on ultimate stress of pure SA-CMC.” 

“Young's modulus of materials is the ratio of stress to strain within the elastic limit. Young's modulus represents the stiffness of the material, the larger Young's modulus, the harder to deform. Young's modulus of pure SA-CMC and SiQDs/SA-CMC with SiQDs loadings of 1, 2 and 3 ml are 0.0125, 0.0055, 0.0053 and 0.0036 MPa, respectively. The results indicate that the Young's modulus of nanocomposites decrease with the addition of SiQDs.” 

“Strain at break is also named elongation, which is utilized to characterize the toughness or brittleness of polymer materials. The strain at break of SA-CMC and SiQDs/SA-CMC with SiQDs loadings of 1, 2 and 3 ml is 300%, 325%, 350% and 390%, respectively. The increase of strain at break demonstrates that the flexibility of SA-CMC film can be improved through adding SiQDs. ” has been changed into “Figure 8 exhibits strain-stress curves of SA-CMC and SiQDs/SA-CMC. Young's modulus of materials is the ratio of stress to strain within the elastic limit. Young's modulus represents the stiffness of the material, the larger Young's modulus, the harder to deform. Young's modulus of pure SA-CMC and SiQDs/SA-CMC with SiQDs loadings of 6 wt%, 12 wt% and 18 wt% are 0.0125, 0.0055, 0.0053 and 0.0036 MPa, respectively. The results indicate that the stiffness of nanocomposites decrease with the addition of SiQDs.

Strain at break is also named elongation, which is utilized to characterize the ductility and toughness of polymer materials. The strain at break of SA-CMC and SiQDs/SA-CMC with SiQDs loadings of 6 wt%, 12 wt% and 18 wt% is 300%, 325%, 350% and 390%, respectively. The increase of strain at break demonstrates that the toughness of SA-CMC film can be improved through adding SiQDs. The ultimate stress of pure SA-CMC and SiQDs/SA-CMC with SiQDs loadings of 6 wt%, 12 wt% and 18 wt% is 1.64, 1.40, 1.60 and 1.67 MPa, respectively, the data above changes little.

The fracture energy is the area under stress-strain curve. It could be also indicated the toughness of polymer materials. The area under stress-strain curve of SA-CMC and SiQDs/SA-CMC with SiQDs loadings of 6 wt%, 12 wt% and 18 wt% is 300, 348, 415 and 487 N/m, respectively. The increase of fracture energy under stress-strain curve indicates that the toughness of SA-CMC film can be improved through adding SiQDs. Furthermore, the higher ultimate stress or strain at break, the greater fracture energy, thus it is consistent with the situation where the elongation at break increases and ultimate stress changes little in this experiment.

It can be interpreted that the decreasing Young modulus, increasing strain at break and fracture energy, and improving toughness of SA-CMC composites with the addition of SiQDs, because the interfacial adhesion is better and the effective interface enhances between SiQDs and the SA-CMC matrix when SiQDs was added into SA-CMC composites. Meanwhile, due to the existent Si-O-Si bonds in SiQDs, sols are formed in solution easily, and the sol in water can interact with SA-CMC to form a colloidal particles penetrating SA-CMC network structure.” in the revised manuscript. (see lines 220-244)  

In addition, we would correct stress-strain curve, as shown in figure 8.

Figure 8. Strain-Stress curves of pure SA-CMC and SiQDs/SA-CMC nanocomposites

Q2-5: It would be good that, based on the properties of nanocomposites, authors could give some ideas where this type of materials could be used.

A2-5. Answer to reviewer: 

We appreciate your question very much.

Aggregation and high cost of SiQDs can result in limitedly optoelectronic application and slow development, therefore, we would display the application where this type of materials could be used with respect to existent literature: “The SiQDs/SA-CMC nanocomposites are promising in optoelectronic application, e.g. light emitting diodes have been reported recently .” (see lines 55-56)

The relevant literature is as follows:

 Wu, Y.S.; Zhang, H.; Pan, A.Z.; Wang, Q.; Zhang, Y.F.; Zhou, G.J.; He, L. White-Light-Emitting Melamine-Formaldehyde Microspheres through Polymer-Mediated Aggregation and Encapsulation of Graphene Quantum Dots. Adv. Sci. 2019, 6.(see Literature [32])

As for the responses to Reviewer 2, it should be mentioned that the revised words are in green color in the revised manuscript. 

Reviewer 3 Report

The paper reports on the synthesis of silicon quantum dots (SiQDs) via one-pot hydrothermal route and properties of obtained SiQDs, native or incorporated in alginate-carboxymethylcellulose composite films. The topic is important and could be of interest for a large scientific community. However, the description of the procedures and methods used need to be improved and the discussion on the results to be enhanced.

In my opinion, the manuscript could be accepted for publication after major revision if the important issues below are addressed in the revised version:

In general, the Introduction should be considerably expanded and improved. The discussion on the up-to date synthetic methods for obtaining quantum dots should be exhausted; the included examples should be critically analyzed and compared. Further in the Introduction, the discussion on the composites and especially those containing quantum dot is also deficient and needs extension and critical analysis. On this basis the novelty of the synthetic strategy proposed in present work should be clearly outline. In Materials and Methods, 4. Characterization Techniques the analytical equipment used in the investigation is described although very briefly. Sample preparation for different analyses and testing, however, is not approached at all. The detailed description of the samples treatment should be carefully described in the manuscript – it is very important not only for the current investigation but also in order to be able to correlate the experimental results with those reported in the literature. In Results and Discussion, 3.6. Formation mechanism of SiQDs (page 6), the proposed mechanism of SiQDs growth, illustrated on Figure 6 is not clear and needs to be elucidated. On the Figure: what is the difference between pathway 2 and pathway 3 where the reagents are identical? What is the meaning of “hydrophilic group”? The manuscript needs thorough English language and style editing

More comments and questions to be considered:

Page 2; lines 60-65: 2.2. Preparation of SiQDs  

            Please describe in details the synthetic procedure: in air or in inert atmosphere? stirring during the heating? “the prepared sample was naturally cooled and ultra-filtered” - Please provide information on the device and characteristics of the (eventual) membrane.

Page 2; lines 66-71: 2.3. Fabrication of SiQDs/SA-CMC Nanocomposites

            “SiQDs with 1, 2 and 3 mL mixed solution” – what is the solvent (solvents) used and concentration?

            “nanocomposite films were labeled as1 mL-SiQDs/SA-CMC, 2 mL-SiQDs/SA-CMC and 3 mL-SiQDs/SA-CMC” – using ml in the labels is not appropriate, better use the weight % of the SiQDs in the composite instead.

Page 2; lines 85-86: “To obtain optimally synthetic conditions of SiQDs, the ratio of the reactants, reaction time and temperature are investigated.”

            What is the influence of concentration of the reagents?

Page 3, lines 88-89: When the molar ratio of NaBH4 to H2SiO3 is from 0.3 to 0.9, the change in fluorescence intensity of SiQDs is relatively diminutive.”

            In fact, for molar ratio 0.3-0.9 the fluorescence intensity increases from 100 to about 250 a.u., so could not be diminutive.

Page 5, lines 157-170: 3.5. Fluorescence stability Analysis of SiQDs

            Please clarify the following expressions: with a total of 7 groups”; “modified SiQDs”; “unmodified SiQDs”.

            How is the statement: “Furthermore, the long-lifetime fluorescence of SiQDs was easily distinguished from the autofluorescence of the living organism, whose lifetimes were less than 5 ns [9].” related with the present work?

Page 7, lines 212-2013: “The increase of strain at break demonstrates that the flexibility of SA-CMC film can be improved through adding of SiQDs.”

            In general, inorganic fillers are known to impart rigidity rather than flexibility in polymer materials. How could you explain the improved strain at break of the SiQDs nanocomposites compared to the neat polymer composition.

Page 8, lines 217-218: The statement in the Conclusions: “the emission spectrum has strong dependence on the excitation wavelength” is not supported by experimental data in the manuscript as the presented results are obtained at λex = 365 nm only.

Author Response

Response to Reviewer 3 comments

Firstly I would like to take this opportunity to thank you for your carefully reading and your constructive comments and suggestions.

Answer to Reviewer #3:

Q3-1: In general, the Introduction should be considerably expanded and improved. The discussion on the up-to date synthetic methods for obtaining quantum dots should be exhausted; the included examples should be critically analyzed and compared.

A3-1. Answer to reviewer: 

Thank you for your suggestion.

We expand and improve the introduction on the up-to date synthetic methods for obtaining quantum dots. New content “Although, preparation methods of SiQDs have been extensively explored with electrochemical etching silicon wafers using HF [15], reduction of hydrogen silsesquioxane by H2 at high temperature [16], laser pyrolysis of silanes [17], microwave assisted method for reducing silane coupling agent by sodium citrate [18], gas phase reduction method for reducing SiO2 gel by H2 or Mg vapor [19], liquid phase reduction silane method for lithium aluminum hydride reduction halosilane [20], high energy grinding ball method based on silicon wafer [21]and so on, these synthetic methods possess complicated procedures, hazardous reagents (HF, H2, H2SO4, SiX4), complex morphology, low purity, uneven size and highly demanding equipment.” was added in the revised manuscript. (see lines 32-40)

The relevant literatures added are as follows:

18.Wang, J.J.; Lian, M.; Xiong, J. A new water-soluble imidazole based silicon quantum dot was prepared and used for fluorescence detection of trace copper in fruits and vegetables. J. Chin. Anal. Chem. 2016, 44, 367. (see Literature [18])

19.Anderson, I.E.; Shircliff, R.A.; Macauley, C. Silanization of Low-Temperature-Plasma Synthesized Silicon Quantum Dots for Production of a Tunable, Stable, Colloidal Solution. J. Phys. Chem. C. 2012, 116, 3979. (see Literature [19])

20.Bley, R.A.; Kauzlarich, S.M. A low-temperature solution phase route for the synthesis of silicon nanoclusters. J. Am. Chem. Soc. 1996, 118, 12461.(see Literature [20])

21.Heintz, A.S.; Fink, M.J.; Mitchell, B.S. Mechanochemical synthesis of blue luminescent alkyl/alkenyl-passivated silicon nanoparticles. Adv. Mater. 2007, 19, 3984. (see Literature [21])

Q3-2: Further in the Introduction, the discussion on the composites and especially those containing quantum dot is also deficient and needs extension and critical analysis. On this basis the novelty of the synthetic strategy proposed in present work should be clearly outline. In Materials and Methods.

A3-2. Answer to reviewer: 

Thank you for your suggestion.

The incorporation of QDs in a transparent polymer matrix is one of the main approaches for their utilization in numerous photonic and optoelectronic applications and integration in real devices. In addition to serving the role of the matrix, polymers provide mechanical and chemical stability to the nanocomposite. Additionally, the introduction of polymers can slow QD agglomeration, thereby maintaining their emission properties. We have changed the content in the revised manuscript .“Recently, many composites have been reported and have attracted considerable attention, in particular, the introduction of polymers not only provide mechanical and chemical stability to the nanocomposite, but slow QD agglomeration to maintain their emission properties [24,25]. In the presenting work, numerous poly(vinyl alcohol) (PVA)/GQD and carboxymethyl chitosan/ZnOQD nanocomposite films were fabricated, these nanocomposite films exhibit  toilless biodegradability, excellent biocompatibility, edibility and non-toxicity, however, poor toughness and limited fluorescence are shortcomings in medical and optoelectronic applications [26,27]. Sodium alginate (SA) and carboxymethylcellulose sodium (CMC) were chosen as matrix polymers because of high optical transparency, chemical stability, film-forming properties, abundant and sustainable sources [28,29]. Additionally, one of the most attractive methods for QDs with polymers is simple solution casting.”

The previous content “Recently, many composites have been reported and have attracted considerable attention [20,21]. In particular, many natural bio-polymers have been widely used in medical applications [22,23] and food packaging [24,25] owing to toilless biodegradability [26], excellent biocompatibility [27], edibility [28] and non-toxicity [29]. Sodium alginate (SA) and carboxymethylcellulose sodium (CMC) as bio-polymers exhibit high light transmittance, chemical stability and film-forming properties [30,31]. Besides, SA and CMC are abundant sources and sustainable. Therefore, in this work, SiQDs were synthesized via one-pot hydrothermal method with ethanol as reagent, silicic acid and sodium borohydride as new raw materials. Furthermore, to obtain a luminescent SA-CMC hybrid film, SiQDs were successfully introduced into SA-CMC polymer matrix through a simple procedure. The SiQDs/SA-CMC nanocomposites are promising in photonic and optoelectronic application [32].” was changed into “Recently, many composites have been reported and have attracted considerable attention, in particular, the introduction of polymers not only provide mechanical and chemical stability to the nanocomposite, but slow QD agglomeration to maintain their emission properties [24,25]. In the presenting work, numerous poly(vinyl alcohol) (PVA)/GQD and carboxymethyl chitosan/ZnOQD nanocomposite films were fabricated, these nanocomposite films exhibit  toilless biodegradability, excellent biocompatibility, edibility and non-toxicity, however, poor toughness and limited fluorescence are shortcomings in medical and optoelectronic applications [26,27]. Sodium alginate (SA) and carboxymethylcellulose sodium (CMC) were chosen as matrix polymers because of high optical transparency, chemical stability, film-forming properties, abundant and sustainable sources [28,29]. Additionally, one of the most attractive methods for QDs with polymers is simple solution casting.” (see lines 42-52)

The relevant literatures added are as follows:

26.Kovalchuk, A.; Huang, K.; Xiang, C.; Marti, A.A.; Tour, J.M. Luminescent Polymer Composite Films Containing Coal-Derived Graphene Quantum Dots. Acs Appl. Mater. Interfaces. 2015, 7, 26063-26068. (see Literature [26])

27.Li, T.; Li, B.; Ji, Y.; Wang, L. Luminescent and UV-Shielding ZnO Quantum Dots/Carboxymethylcellulose Sodium Nanocomposite Polymer Films. Polymers. 2019, 10, 1112-1119. (see Literature [27])

The relevant literatures deleted are as follows:

23.Hu, H.; Shang, C.H.; Wang, J.J. Medical Image Analysis of Bacteria Presence on Nickel-Phosphorus Based Nanocomposite Film Applied to Health Informatics. RSC Adv. 2018, 8, 1121-1125. (see Previous literature [23])

24.Bibi, F.; Guillaume, C.; Gontard, N.; Sorli, B. Wheat gluten, a bio-polymer to monitor carbon dioxide in food packaging: Electric and dielectric characterization. Sens. Actuators, B. 2017, 250, 76-84. (see Previous literature [24])

25.Li, K.; Jin, S.; Liu, X.; Chen, H.; He, J.; Li, J. Preparation and Characterization of Chitosan/Soy Protein Isolate Nanocomposite Film Reinforced by Cu Nanoclusters. Polymers. 2017, 9, 247. (see Previous literature [25])

26.Xu, W.; Xiao, M.; Yuan, L.; Zhang, J.; Hou, Z. Preparation, Physicochemical Properties and Hemocompatibility of Biodegradable Chitooligosaccharide-Based Polyurethane. Polymers. 2018, 10, 580. (see Previous literature [26])

27.Zhong, Q.; Tian, J.; Liu, T.; Guo, Z.; Ding, S.; Li, H. Preparation and antibacterial properties of carboxymethyl chitosan/ZnO nanocomposite microspheres with enhanced biocompatibility. Mater. Lett. 2018, 212, 58-61. (see Previous literature [27])

28.Bibi, F.; Guillaume, C.; Vena, A.; Gontard, N.; Sorli, B. Wheat gluten, a bio-polymer layer to monitor relative humidity in food packaging: Electric and dielectric characterization. Sens. Actuators, A. 2016, 247, 355-367. (see Previous literature [28])

29.Caro, N.; Medina, E.; Díaz-Dosque, M.; López, L.; Abugoch, L.; Tapia, C. Novel active packaging based on films of chitosan and chitosan/quinoa protein printed with chitosan-tripolyphosphate-thymol nanoparticles via thermal ink-jet printing. Food Hydrocoll. 2016, 52, 520-532. (see Previous literature [29])

Q3-3: Characterization Techniques the analytical equipment used in the investigation is described although very briefly. Sample preparation for different analyses and testing, however, is not approached at all. The detailed description of the samples treatment should be carefully described in the manuscript – it is very important not only for the current investigation but also in order to be able to correlate the experimental results with those reported in the literature.

A3-3. Answer to reviewer: 

We appreciate your question very much.

The previous content “The fourier transform infrared (FTIR) spectroscopy was determined by an Avatar 360 Fourier transform infrared spectrometer (Thermo Nicolet Co., USA). The X-ray photoelectron spectroscopy (XPS) was tested with THERMO-X-ray photoelectron spectrometer (Thermo Electron Co., USA). The photoluminescence spectra (PL) of SiQDs and SiQDs/SA-CMC nanocomposites were recorded on a LS55 fluorescence spectrophotometer (Perkin Elmer Co., USA), respectively. PL lifetime decay of SiQDs was examined with F-7000 fluorescence spectrophotometer (Japan). ” has been changed into “The fourier transform infrared (FTIR) spectroscopy was determined by an Avatar 360 Fourier transform infrared spectrometer (Thermo Nicolet Co., USA) using air as background in three scan times and a scan range between 400 cm-1 and 4000 cm-1. The X-ray photoelectron spectroscopy (XPS) was tested with THERMO-X-ray photoelectron spectrometer (Thermo Electron Co., USA). The photoluminescence spectra (PL) of SiQDs (the concentration in solution was 20%) and SiQDs/SA-CMC nanocomposites were recorded on a LS55 fluorescence spectrophotometer (Perkin Elmer Co., USA) using 365 nm excitation in a scan speed of 500 nm/s and slit width of 8 nm, respectively. PL lifetime decay of SiQDs (the concentration in solution was 20%) was examined with F-7000 fluorescence spectrophotometer (Japan). The strain-stress curves were tested with LRX-PLUS universal material testing machine (Aiteng Co., China) using a 8 cm x 1 cm sample at a scan speed of 500 mm/min.” in the revised manuscript. (see lines 89-100)

Q3-4: In Results and Discussion, 3.6. Formation mechanism of SiQDs (page 6), the proposed mechanism of SiQDs growth, illustrated on Figure 6 is not clear and needs to be elucidated. On the Figure: what is the difference between pathway 2 and pathway 3 where the reagents are identical? What is the meaning of “hydrophilic group”? The manuscript needs thorough English language and style editing.

A3-4. Answer to reviewer: 

Thank you for your question.

We would rewrite the formation mechanism of SiQDs and repaint mechanism of SiQDs with respect to existent literature. The previous content “The growth mechanism of SiQDs is explained, as shown in Figure 6. Because H2SiO3 is not easily hydrolyzed to form silanol, NaBH4 reduces H2SiO3 to form Si-Si bond during hydrothermal heating. Under high temperature and pressure, the Si-Si bond is broken and the silicon alcohol is formed in ethanol. Some silicon alcohols are deoxidized by NaBH4 to form Si-O-Si core of network structure, some silicon alcohols to form Si-C core, and a few remain to form Si core. The above nuclei undergo Ostwald ripening process, and the crystals grow gradually to form stable SiQDs [19].” has been changed into “Through above analysis of FTIR and XPS, SiQDs possess Si-O-Si bond, Si-Si bond and Si-C bond, thus the growth mechanism of SiQDs is explained by these bonds, as shown in Figure 6. In the hydrothermal process, the growth of SiQDs follows bottom-up synthesis mechanism, that is, the bond first forms a core in hydrophilic groups such as -OH, and then the core forms a QD via Ostwald ripening process [23]. Under high temperature and pressure, Si-O-Si bonds, Si-Si bonds and Si-C bonds first form Si-O-Si cores, Si cores and Si-C cores in hydrophilic environment (-OH groups of ethanol). And then above cores grow gradually to form steady SiQDs undergo Ostwald ripening process, respectively.” in the revised manuscript. (see lines 194-201)

Q3-5: Page 2; lines 60-65: 2.2. Preparation of SiQDs. Please describe in details the synthetic procedure: in air or in inert atmosphere? stirring during the heating? “the prepared sample was naturally cooled and ultra-filtered” - Please provide information on the device and characteristics of the (eventual) membrane.

A3-5. Answer to reviewer: 

Thank you for your question.

The previous content “Firstly, silicic acid of 0.39 g (5 mmol) and ethanol (8 mL) was put into reactor. Sodium borohydride of 0.227 g (6 mmol) was slowly added to the above system under stirring for 10 min. Secondly, the reactor was placed in an oven at 200 °C for 3 h. Thirdly, the prepared sample was naturally cooled and ultra-filtered to obtain a light yellow solution. Then, resulting silicon quantum dots were dried in the oven at 40 °C and were labeled as SiQDs.” has been changed into “Firstly, silicic acid of 0.39 g (5 mmol) and ethanol (8 mL) was put into a reactor. Sodium borohydride of 0.227 g (6 mmol) was slowly added to the above system under stirring for 10 min during normal temperature and pressure. Secondly, the reactor was placed in an oven at 200 °C for 3 h. Thirdly, the prepared sample was naturally cooled and ultra-filtered (aperture: 0.22 μm) to obtain a light yellow solution. Then, resulting silicon quantum dots were dried in the oven at 40 °C and were labeled as SiQDs.” in the revised manuscript. (see lines 70-75)

In addition, we added the following content “Disposable needle filter (13*0.22 μm) was purchased from Jiangsu Green Union Science Instrument Co. Ltd. (Jiangsu, China)” in the revised manuscript. (see lines 67-68)

Q3-6: Page 2; lines 66-71: 2.3. Fabrication of SiQDs/SA-CMC Nanocomposites“SiQDs with 1, 2 and 3 mL mixed solution” – what is the solvent (solvents) used and concentration?

“nanocomposite films were labeled as 1 mL-SiQDs/SA-CMC, 2 mL-SiQDs/SA-CMC and 3 mL-SiQDs/SA-CMC” – using ml in the labels is not appropriate, better use the weight % of the SiQDs in the composite instead.

A3-6. Answer to reviewer: 

We appreciate your suggestion very much.

1) The previous content “Then SA-CMC of 10 mL and SiQDs with 1, 2 and 3 mL mixed solution were cast onto glass substrates, respectively. The above SiQDs/SA-CMC nanocomposites transferred to the oven and dried at 45 °C for 12 h. The obtained SiQDs/SA-CMC nanocomposite films were labeled as 1 mL-SiQDs/SA-CMC, 2 mL-SiQDs/SA-CMC and 3 mL-SiQDs/SA-CMC, respectively.” has been changed into “Then SiQDs and SA-CMC aqueous solution were mixed. The mixtures were cast onto glass substrates with no post treatment. Furthermore, the glass substrates were transferred to the oven and dried for 12 h at 45 °C to obtain SiQDs/SA-CMC films (40 square centimeters). The SiQDs/SA-CMC composites with 6, 12 and 18 wt% loading of SiQDs were fabricated, respectively. Thus, the resulting SiQDs/SA-CMC films were labeled as 6 wt% SiQDs/SA-CMC,12 wt% SiQDs/SA-CMC and 18 wt% SiQDs/SA-CMC, respectively.” in the revised manuscript. (see lines 78-84);

The previous content“Figure 7a shows the PL spectra of SA-CMC nanocomposites with SiQDs loadings of 1, 2 and 3 ml at 365 nm excitation wavelength.” has been changed into “Figure 7a shows the PL spectra of SA-CMC nanocomposites with SiQDs loadings of 6 wt%, 12 wt% and 18 wt% at 365 nm excitation wavelength.” in the revised manuscript. (see lines 204-205);

3) The previous content “Young's modulus of pure SA-CMC and SiQDs/SA-CMC with SiQDs loadings of 1, 2 and 3 ml are 0.0125, 0.0055, 0.0090 and 0.0036 MPa, respectively.” has been changed into “Young's modulus of pure SA-CMC and SiQDs/SA-CMC with SiQDs loadings of 6 wt%, 12 wt% and 18 wt% are 0.0125, 0.0055, 0.0053 and 0.0036 MPa, respectively.” in the revised manuscript. (see lines 223, 228 and 234).

Q3-7: Page 2; lines 85-86: “To obtain optimally synthetic conditions of SiQDs, the ratio of the reactants, reaction time and temperature are investigated.” What is the influence of concentration of the reagents?

A3-7. Answer to reviewer: 

We appreciate your question very much.

In this work, SiQDs were synthesized via one-pot hydrothermal method with ethanol as reagent, silicic acid and sodium borohydride as new raw materials. The influence of concentration of the ethanol reagent has little significance, and it is similar to water as a reagent. In comparison, the influence of the ratio of the reactants reaction time and temperature is more obvious.

Q3-8: Page 3, lines 88-89: “When the molar ratio of NaBH4 to H2SiO3 is from 0.3 to 0.9, the change in fluorescence intensity of SiQDs is relatively diminutive.” In fact, for molar ratio 0.3-0.9 the fluorescence intensity increases from 100 to about 250 a.u., so could not be diminutive.

A3-8. Answer to reviewer: 

We appreciate your suggestion very much, it is my honor to be pointed out the writing error, I would correct it.

The previous content “When the molar ratio of NaBH4 to H2SiO3 is from 0.3 to 0.9, the change in fluorescence intensity of SiQDs is relatively diminutive. ” has been changed into “As the molar ratio of NaBH4 to H2SiO3 is from 0.3 to 0.9, the change in fluorescence intensity of SiQDs is increasing, as the molar ratio of NaBH4 to H2SiO3 increases to 1.2:1, the fluorescence intensity is maximum. However, with the molar ratio of NaBH4 to H2SiO3 is from 1.5 to 3.0, the fluorescence intensity weakens. ” in the revised manuscript. (see lines 107-110)  

Q3-9: Page 5, lines 157-170: 3.5. Fluorescence stability Analysis of SiQDs

Please clarify the following expressions: ” with a total of 7 groups”; “modified SiQDs”; “unmodified SiQDs”. 

A3-9. Answer to reviewer: 

Thank you for your question. 

The previous content “with a total of 7 groups” has been changed into “with a total of 35 days” in the revised manuscript. (see line179)

The existent literature confirmed that the surface-unmodified QDs easily agglomerate to result in fluorescent intensity decreasingly, and the surface-modified QDs possess fluorescent stability. During the XPS and FTIR measurements in this paper, it is speculated that there are two forms of QDs. Thus, fluorescence stability measurement are explained above. We would correct “modified SiQDs” and “unmodified SiQDs” to “surface-modified SiQDs” and “surface-unmodified SiQDs” in the revised manuscript. (see lines181-182)  

The relevant literature is as follows:

34.Rodio, M.; Brescia, R.; Diaspro, A.; Intartaglia, R. Direct surface modification of ligand-free silicon quantum dots prepared by femtosecond laser ablation in deionized water. J. Colloid Interface Sci. 2016, 465, 242-248. (see Literature [34])

Q3-10: How is the statement: “Furthermore, the long-lifetime fluorescence of SiQDs was easily distinguished from the autofluorescence of the living organism, whose lifetimes were less than 5 ns.” related with the present work?

A3-10. Answer to reviewer: 

Thank you for your question.

As for “Furthermore, the long-lifetime fluorescence of SiQDs was easily distinguished from the autofluorescence of the living organism, whose lifetimes were less than 5 ns.” We have clarified the expression and added new content: “Such SiQDs are not only promising in optoelectronic application, but significance of imaging application in vitro and vivo” in this section. (see lines190-191)  

The relevant literature is as follows:

9.Ye, H.L.; Cai, S.J.; Li, S.; He, X.W.; Li, W.Y.; Li, Y.H.; Zhang, Y.K. One-Pot Microwave Synthesis of Water-Dispersible, High Fluorescence Silicon Nanoparticles and Their Imaging Applications in Vitro and in Vivo. Anal. Chem. 2016, 88, 11631-11638. (see Literature [9])

Q3-11: Page 7, lines 212-2013: “The increase of strain at break demonstrates that the flexibility of SA-CMC film can be improved through adding of SiQDs.”In general, inorganic fillers are known to impart rigidity rather than flexibility in polymer materials. How could you explain the improved strain at break of the SiQDs nanocomposites compared to the neat polymer composition?

A3-11. Answer to reviewer: 

We appreciate your suggestion very much.

We have added the content of fracture energy and explain the reason that Young modulus decreases and elongation at break increase with the addition of SiQDs in the revised manuscript. “The fracture energy is the area under stress-strain curve. It could be also indicated the toughness of polymer materials. The area under stress-strain curve of SA-CMC and SiQDs/SA-CMC with SiQDs loadings of 6 wt%, 12 wt% and 18 wt% is 300, 348, 415 and 487 N/m, respectively. The increase of fracture energy under stress-strain curve indicates that the toughness of SA-CMC film can be improved through adding SiQDs. Furthermore, the higher ultimate stress or strain at break, the greater fracture energy, thus it is consistent with the situation where the elongation at break increases and ultimate stress changes little in this experiment.” 

“It can be interpreted that the decreasing Young modulus, increasing strain at break and fracture energy, and improving toughness of SA-CMC composites with the addition of SiQDs, because the interfacial adhesion is better and the effective interface enhances between SiQDs and the SA-CMC matrix when SiQDs was added into SA-CMC composites. Meanwhile, due to the existent Si-O-Si bonds in SiQDs, sols are formed in solution easily, and the sol in water can interact with SA-CMC to form a colloidal particles penetrating SA-CMC network structure.”

The previous content “Figure 8 exhibits strain-stress curves of SA-CMC and SiQDs/SA-CMC. The ultimate stress of pure SA-CMC and SiQDs/SA-CMC with SiQDs loadings of 1, 2 and 3 ml is 1.64, 1.40, 1.60 and 1.67 MPa, respectively. In comparison with pure SA-CMC film, the ultimate stress of 1 mL-SiQDs/SA-CMC and 2 mL-SiQDs/SA-CMC reduces and 3 mL-SiQDs/SA-CMC slightly increases. Ultimate stress reflects the fracture resistance of materials, the larger ultimate stress, the stiffer material. The data above exhibits quite small, therefore, it can be confirmed that SiQDs/SA-CMC films are ductile material and SiQDs has little influence on ultimate stress of pure SA-CMC.”

“Young's modulus of materials is the ratio of stress to strain within the elastic limit. Young's modulus represents the stiffness of the material, the larger Young's modulus, the harder to deform. Young's modulus of pure SA-CMC and SiQDs/SA-CMC with SiQDs loadings of 1, 2 and 3 ml are 0.0125, 0.0055, 0.0053 and 0.0036 MPa, respectively. The results indicate that the Young's modulus of nanocomposites decrease with the addition of SiQDs.”

“Strain at break is also named elongation, which is utilized to characterize the toughness or brittleness of polymer materials. The strain at break of SA-CMC and SiQDs/SA-CMC with SiQDs loadings of 1, 2 and 3 ml is 300%, 325%, 350% and 390%, respectively. The increase of strain at break demonstrates that the flexibility of SA-CMC film can be improved through adding of SiQDs. ” has been changed into “Figure 8 exhibits strain-stress curves of SA-CMC and SiQDs/SA-CMC. Young's modulus of materials is the ratio of stress to strain within the elastic limit. Young's modulus represents the stiffness of the material, the larger Young's modulus, the harder to deform. Young's modulus of pure SA-CMC and SiQDs/SA-CMC with SiQDs loadings of 6 wt%, 12 wt% and 18 wt% are 0.0125, 0.0055, 0.0053 and 0.0036 MPa, respectively. The results indicate that the stiffness of nanocomposites decrease with the addition of SiQDs.

Strain at break is also named elongation, which is utilized to characterize the ductility and toughness of polymer materials. The strain at break of SA-CMC and SiQDs/SA-CMC with SiQDs loadings of 6 wt%, 12 wt% and 18 wt% is 300%, 325%, 350% and 390%, respectively. The increase of strain at break demonstrates that the toughness of SA-CMC film can be improved through adding SiQDs. The ultimate stress of pure SA-CMC and SiQDs/SA-CMC with SiQDs loadings of 6 wt%, 12 wt% and 18 wt% is 1.64, 1.40, 1.60 and 1.67 MPa, respectively, the data above changes little.

The fracture energy is area under stress-strain curve, it could be also indicated the toughness of polymer materials. The area under stress-strain curve of SA-CMC and SiQDs/SA-CMC with SiQDs loadings of 6 wt%, 12 wt% and 18 wt% is 300, 348, 415 and 487 N/m, respectively. The increase of fracture energy under stress-strain curve indicates that the toughness of SA-CMC film can be improved through adding SiQDs. Furthermore, the higher ultimate stress or strain at break, the greater fracture energy, thus it is consistent with the situation where the elongation at break increases and ultimate stress changes little in this experiment.

It can be interpreted that the decreasing Young modulus, increasing strain at break and fracture energy, and improving toughness of SA-CMC composites with the addition of SiQDs, because the interfacial adhesion is better and the effective interface enhances between SiQDs and the SA-CMC matrix when SiQDs was added into SA-CMC composites. Meanwhile, due to the existent Si-O-Si bonds in SiQDs, sols are formed in solution easily, and the sol in water can interact with SA-CMC to form a colloidal particles penetrating SA-CMC network structure.” in the revised manuscript. (see lines 220-244)  

Q3-12: Page 8, lines 217-218: The statement in the Conclusions: “the emission spectrum has strong dependence on the excitation wavelength” is not supported by experimental data in the manuscript as the presented results are obtained at λex = 365 nm only.  

A3-12. Answer to reviewer: 

We appreciate your question very much.

In the manuscript, we have displayed the experimental data in Figure 1d, it shows the fluorescence spectra of SiQDs at different excitation wavelengths: “When the excitation wavelength increases from 355 to 425 nm, the fluorescence intensity of SiQDs show the gradual decrease and red shift of the emission wavelength. The result indicates that the emission wavelength of SiQDs has a strong dependence on the excitation wavelength. Figure 1d confirmed that the emission peak ranges from 400 to 500 nm, belonging to the blue light range.” (see lines131-137)

Figure 1 (d): different excitation wavelengths (λex=355-425 nm)

Q3-13: The manuscript needs thorough English language and style editing.

A3-13. Answer to reviewer: 

Thank you for your suggestion. English language and style are carefully edited and corrected in the revised manuscript. In addition, we consulted a professional teacher to help us polish our English.

As for the responses to Reviewer 3, it should be mentioned that the revised words are in purple color in the revised manuscript. 

Round 2

Reviewer 3 Report

The authors made efforts to improve the manuscript taking into account the questions and comments raised. The discussion is enhanced and the contribution of the work the field is better outlined.

Still some of the statements and explanations are difficult to understand obviously due to the wrong terms and/or English expressions used. For example:

 line 48: “In the presenting work, numerous poly(vinyl alcohol) (PVA)/GQD and carboxymethyl chitosan/ZnOQD nanocomposite films were fabricated, these nanocomposite films exhibit toilless biodegradability, excellent biocompatibility, edibility and non-toxicity, however, poor toughness and limited  fluorescence are shortcomings in medical and optoelectronic applications [26,27].” – it is unclear if the composites in question are prepared by the authors or these is information from literature  search

line 55: “Additionally, one of the most attractive methods for QDs with polymers is simple solution  casting.” – unclear sentence

lines 85-136: In 2.4.Characterization Techniques section, the description of the sample preparation for analysis is still incomplete or missing. For example: is FTIR spectroscopy performed on neat sample, in solution (in what solvent) or in KBr?

“photoluminescence spectra (PL) of SiQDs (the concentration in solution was 20%)” – which is the solvent?

“The strain-stress curves were tested with 134 LRX-PLUS universal material testing machine (Aiteng Co., China) using a 8 cm × 1 cm sample…” – what is the sample thickness?

lines 231-232: The comment concerning the terms “modified SiQDs” and “unmodified SiQDs” is addressed, but introduced terms “surface-modified ‘”/“surface-unmodified SiQDs”  in the in the revised manuscript does not clarify the issue: is there any intentional chemical modification carried out? If not, the use of above terms is improper.

Author Response

Response to Reviewer 3 comments

Firstly I would like to take this opportunity to thank you for your carefully reading and your constructive comments and suggestions again.

Answer to Reviewer #3:

Q3-1: line 48: “In the presenting work, numerous poly(vinyl alcohol) (PVA)/GQD and carboxymethyl chitosan/ZnOQD nanocomposite films were fabricated, these nanocomposite films exhibit toilless biodegradability, excellent biocompatibility, edibility and non-toxicity, however, poor toughness and limited  fluorescence are shortcomings in medical and optoelectronic applications [26,27].” – it is unclear if the composites in question are prepared by the authors or these is information from literature  search

A3-1. Answer to reviewer: 

We appreciate your question very much. The composites is from previous literature, we corrected it clearly in the revised manuscript. The previous content “In the presenting work, numerous poly(vinyl alcohol) (PVA)/GQD and carboxymethyl chitosan/ZnOQD nano-composite films were fabricated, these nanocomposite films exhibit toilless biodegradability, excellent biocompatibility, edibility and non-toxicity, however, poor toughness and limited  fluorescence are shortcomings in medical and optoelectronic applications [26,27].” has been changed into “In the reported literature, poly(vinyl alcohol) (PVA)/GQD and carboxymethyl chitosan/ZnOQD nanocomposite films exhibit toilless biodegradability, excellent biocompatibility, edibility and non-toxicity [26,27].” in the revised manuscript. (see lines 45-47)

Q3-2: line 55: “Additionally, one of the most attractive methods for QDs with polymers is simple solution  casting.” – unclear sentence

A3-2. Answer to reviewer: 

We appreciate your question very much. We rewrote this sentence in the revised manuscript. The previous content “Additionally, one of the most attractive methods for QDs with polymers is simple solution casting. Therefore, in this work, SiQDs were synthesized via one-pot hydrothermal method with ethanol as reagent, silicic acid and sodium borohydride as new raw materials. Furthermore, to obtain a luminescent SA-CMC hybrid film, SiQDs were successfully introduced into SA-CMC polymer matrix through a simple procedure. The SiQDs/SA-CMC nanocomposites are promising in optoelectronic application, e.g. light emitting diodes have been reported recently [30].” has been changed into “Therefore, in this work, SiQDs were synthesized via one-pot hydrothermal method with ethanol as reagent, silicic acid and sodium borohydride as new raw materials. Furthermore, to obtain a luminescent SA-CMC hybrid film, SiQDs and SA-CMC mixed polymer matrix are poured on the glass plate, and the film is dried after static defoaming. The obtained SiQDs/SA-CMC nanocomposites are promising in optoelectronic application, e.g. light emitting diodes have been reported recently [30]. ” in the revised manuscript. (see lines 49-54)

Q3-3: lines 85-136: In 2.4.Characterization Techniques section, the description of the sample preparation for analysis is still incomplete or missing. For example: is FTIR spectroscopy performed on neat sample, in solution (in what solvent) or in KBr?

A3-3. Answer to reviewer: 

We appreciate your question very much. We added the content “in ethanol solvent” in the revised manuscript.

The previous content “The fourier transform infrared (FTIR) spectroscopy was determined by an Avatar 360 Fourier transform infrared spectrometer (Thermo Nicolet Co., USA) using air as background in three scan times and a scan range between 400 cm-1 and 4000 cm-1.” has been changed into “The fourier transform infrared (FTIR) spectroscopy was performed in ethanol solvent by an Avatar 360 Fourier transform infrared spectrometer (Thermo Nicolet Co., USA) using air as background in three scan times and a scan range between 400 cm-1 and 4000 cm-1.” in the revised manuscript. (see lines 87-90)

Q3-4: “photoluminescence spectra (PL) of SiQDs (the concentration in solution was 20%)” – which is the solvent?

A3-4. Answer to reviewer: 

We appreciate your question very much. We corrected it in the revised manuscript. The previous content “photoluminescence spectra (PL) of SiQDs (the concentration in solution was 20%)” has been changed into “photoluminescence spectra (PL) of SiQDs (the concentration in ethanol solvent was 20%)” in the revised manuscript. (see lines 91-92)

Q3-5: “The strain-stress curves were tested with 134 LRX-PLUS universal material testing machine (Aiteng Co., China) using a 8 cm × 1 cm sample…” – what is the sample thickness?

A3-5. Answer to reviewer: 

Thanks for your good suggestion. We added the content “SiQDs/SA-CMC sample (length of 8 cm, width of 1 cm and thickness of 0.05 cm)” in the revised manuscript.

The previous content “The strain-stress curves were tested with LRX-PLUS universal material testing machine (Aiteng Co., China) using a 8 cm × 1 cm sample in a scan speed of 500 mm/min.” has been changed into “The strain-stress curves were tested with LRX-PLUS universal material testing machine (Aiteng Co., China) using SiQDs/SA-CMC sample (length of 8 cm, width of 1 cm and thickness of 0.05 cm) in a scan speed of 500 mm/min.” in the revised manuscript. (see lines 96-98)

Q3-6: lines 231-232: The comment concerning the terms “modified SiQDs” and “unmodified SiQDs” is addressed, but introduced terms “surface-modified ‘”/“surface-unmodified SiQDs”  in the in the revised manuscript does not clarify the issue: is there any intentional chemical modification carried out? If not, the use of above terms is improper.

A3-6. Answer to reviewer: 

We appreciate your question very much. We corrected it on the basis of your good advice.  There is not intentional chemical modification carried out and the use of above terms is improper, therefore, we deleted this description in the revised manuscript.

 The previous content “The fluorescent intensity decreases from 1 to 10 days because of the agglomeration of surface-unmodified SiQDs during the testing process [32]. Furthermore, as the surface-unmodified SiQDs are oxidized or agglomerated, the surface-modified SiQDs remain stable structurally and thus have stable fluorescence after 10 days.” has been changed into “The fluorescent intensity decreases from 1 to 10 days because of the agglomeration of SiQDs during the testing process [32]. However, SiQDs with stable structure remain fluorescent stability after 10 days.” in the revised manuscript. (see lines 176-178)

As for the responses to Reviewer 3, it should be mentioned that the revised words are in blue color in the revised manuscript.

Round 3

Reviewer 3 Report

Considering the changes made in the third version of the manuscript and possibility for additional English language and style editing after the acceptance, I recommend this paper to be accepted for publication in Polymers.